# Prevalence and Risk Factors of Infection in the Representative COVID-19 Cohort Munich

**DOI:** 10.3390/ijerph18073572

**Published:** 2021-03-30

**Authors:** Michael Pritsch, Katja Radon, Abhishek Bakuli, Ronan Le Gleut, Laura Olbrich, Jessica Michelle Guggenbüehl Noller, Elmar Saathoff, Noemi Castelletti, Mercè Garí, Peter Pütz, Yannik Schälte, Turid Frahnow, Roman Wölfel, Camilla Rothe, Michel Pletschette, Dafni Metaxa, Felix Forster, Verena Thiel, Friedrich Rieß, Maximilian Nikolaus Diefenbach, Günter Fröschl, Jan Bruger, Simon Winter, Jonathan Frese, Kerstin Puchinger, Isabel Brand, Inge Kroidl, Jan Hasenauer, Christiane Fuchs, Andreas Wieser, Michael Hoelscher

**Affiliations:** 1Division of Infectious Diseases and Tropical Medicine, University Hospital, LMU Munich, 80802 Munich, Germany; pritsch@lrz.uni-muenchen.de (M.P.); bakuli@lrz.uni-muenchen.de (A.B.); olbrich@lrz.uni-muenchen.de (L.O.); guggenbuehl@lrz.uni-muenchen.de (J.M.G.N.); saathoff@lrz.uni-muenchen.de (E.S.); castelletti@lrz.uni-muenchen.de (N.C.); rothe@lrz.uni-muenchen.de (C.R.); Michel.Pletschette@lrz.uni-muenchen.de (M.P.); metaxa@lrz.uni-muenchen.de (D.M.); Verena.Thiel@lrz.uni-muenchen.de (V.T.); Friedrich.Riess@lrz.uni-muenchen.de (F.R.); diefenbach@lrz.uni-muenchen.de (M.N.D.); froeschl@lrz.uni-muenchen.de (G.F.); jan.bruger@med.uni-muenchen.de (J.B.); simon.winter@med.uni-muenchen.de (S.W.); jonathan.frese@med.uni-muenchen.de (J.F.); kerstin.puchinger@med.uni-muenchen.de (K.P.); isabel.brand@med.uni-muenchen.de (I.B.); ikroidl@lrz.uni-muenchen.de (I.K.); wieser@mvp.lmu.de (A.W.); 2German Center for Infection Research (DZIF), Partner Site Munich, 80802 Munich, Germany; RomanWoelfel@Bundeswehr.org; 3Institute and Outpatient Clinic for Occupational, Social and Environmental Medicine, University Hospital, LMU Munich, 80336 Munich, Germany; katja.radon@med.uni-muenchen.de (K.R.); Felix.Forster@med.uni-muenchen.de (F.F.); 4Center for International Health (CIH), University Hospital, LMU Munich, 80336 Munich, Germany; 5Comprehensive Pneumology Center (CPC) Munich, German Center for Lung Research (DZL), 89337 Munich, Germany; 6Helmholtz Zentrum München—German Research Center for Environmental Health, Institute of Computational Biology, 85764 Neuherberg, Germany; ronan.legleut@helmholtz-muenchen.de (R.L.G.); merce.gari@helmholtz-muenchen.de (M.G.); peter.puetz@uni-bielefeld.de (P.P.); yannik.schaelte@helmholtz-muenchen.de (Y.S.); turid.frahnow@helmholtz-muenchen.de (T.F.); jan.hasenauer@uni-bonn.de (J.H.); christiane.fuchs@helmholtz-muenchen.de (C.F.); 7Helmholtz Zentrum München—German Research Center for Environmental Health, Core Facility Statistical Consulting, 85764 Neuherberg, Germany; 8Faculty of Business Administration and Economics, Bielefeld University, 33615 Bielefeld, Germany; 9Center for Mathematics, Technische Universität München, 85748 Garching, Germany; 10Bundeswehr Institute of Microbiology, 80937 Munich, Germany; 11Interdisciplinary Research Unit Mathematics and Life Sciences, University of Bonn, 53113 Bonn, Germany

**Keywords:** COVID-19, SARS-CoV-2, population-based cohort study, seroprevalence, infection fatality ratio, underreporting

## Abstract

Given the large number of mild or asymptomatic SARS-CoV-2 cases, only population-based studies can provide reliable estimates of the magnitude of the pandemic. We therefore aimed to assess the sero-prevalence of SARS-CoV-2 in the Munich general population after the first wave of the pandemic. For this purpose, we drew a representative sample of 2994 private households and invited household members 14 years and older to complete questionnaires and to provide blood samples. SARS-CoV-2 seropositivity was defined as Roche N pan-Ig ≥ 0.4218. We adjusted the prevalence for the sampling design, sensitivity, and specificity. We investigated risk factors for SARS-CoV-2 seropositivity and geospatial transmission patterns by generalized linear mixed models and permutation tests. Seropositivity for SARS-CoV-2-specific antibodies was 1.82% (95% confidence interval (CI) 1.28–2.37%) as compared to 0.46% PCR-positive cases officially registered in Munich. Loss of the sense of smell or taste was associated with seropositivity (odds ratio (OR) 47.4; 95% CI 7.2–307.0) and infections clustered within households. By this first population-based study on SARS-CoV-2 prevalence in a large German municipality not affected by a superspreading event, we could show that at least one in four cases in private households was reported and known to the health authorities. These results will help authorities to estimate the true burden of disease in the population and to take evidence-based decisions on public health measures.

## 1. Introduction

The COVID-19 pandemic has changed life across the globe. The global case number, based on positive PCR results, is ever increasing. In Germany, the first COVID-19 case was diagnosed in Munich on 27 January 2020 [1]. While further spread could be limited to a cluster of 15 co-workers and their families by thorough contact tracing and quarantine, the next cases were detected only 5 weeks later. Following this, the number of SARS-CoV-2-infected individuals in Munich increased exponentially from 44 cases diagnosed by 7 March to 3304 cases diagnosed by 3 April 2020 [2]. Overall, Germany had the second highest number of registered COVID-19 cases in Europe in late February 2020, surpassed only by Italy. Accordingly, the mitigation stage of the German National Pandemic Plan was implemented, including an 8 week long “lockdown” beginning on March 16 with restrictions such as school, shop, restaurant and hotel closures in 14 of the 16 German federal states, including Bavaria. With these measures, the first wave of the pandemic was considered under control by early May 2020 and a step-by-step relaxation of public health measures followed. At the same time, public health measures such as the use of facemask was made obligatory in public places in Bavaria and other federal states of Germany. The second wave of the pandemic started in Germany in early October 2020. This time, a “lockdown light” was implemented, during which, e.g., schools and shops were left open, resulting in hospitals reaching their capacity limit in several parts of Germany. In Munich, case numbers rose to 8136 active infections registered by the health authorities on 19 December 2020. Schools and shops were hence again closed in all parts of the country from 16 December 2020 to 8 March 2021. In parallel, vaccination started targeting the highest risk group (people over the age of 80 years, residents of homes for the elderly, healthcare workers at highest risk of infection). Starting from 6 March 2021, self-testing was added to the public health measures aiming at the control of the pandemic in Germany [3].

The numbers given above are based on official case reports. However, the number of asymptomatic individuals or patients with mild symptoms not reporting to the health system is largely unknown. This knowledge is crucial to estimate the burden of disease in a population, including true reproduction numbers and attack rates [4]. Up to now, only limited peer-reviewed data from population-based cross-sectional serosurveys on SARS-CoV-2 antibody prevalence at different time points are available. A meta-analysis also evaluating pre-prints of seroprevalence studies [5] reported seroprevalences ranged from below 1% in Iceland to more than 30% in Guilan province, Iran [6,7].

In Germany, one population-based study was conducted following a superspreading event in a smaller town, showing an adjusted immunoglobulin G (IgG) seroprevalence close to 20% [8]. These previous serosurveys used diverse sampling as well as testing methods with varying validity, affecting comparability as well as interpretability of results [3]. Although testing for SARS-CoV-2-specific antibodies seems superior to PCR testing for defining the real dimension of past infections, high sensitivity and especially specificity are crucial due to the low prevalence at the population level [9].

We therefore aimed to identify the complete SARS-CoV-2 seroprevalence in Munich private households including asymptomatic persons and mildly affected patients not reporting to the healthcare system. By doing so, we aimed to provide health authorities with information on the population still at risk for SARS-CoV-2 infections after the first wave of the pandemic. 

## 2. Methods

**Study design, setting, and population.** A detailed description of the study design, setting, and population was previously published [10]; details on sampling design and statistical considerations are given in Online Appendix A. In short, we carried out the fieldwork for this baseline of a future cohort study between 5 April and 12 June 2020, in which we selected a random sample of 100 out of 755 Munich constituencies as starting points to represent the Munich population (Figure 1A). Using random route methodology, fieldworkers selected approximately 30 households per constituency, starting from the city’s geographic center. They partly crossed the borders of constituencies, resulting in a selection of 2994 households in 368 of the 755 constituencies (Figure 1B). If multi-party houses were selected, we aimed to include 1 household per floor to study transmission dynamics within buildings. The mean number of households recruited per house varied between 1 and 7 across constituencies (Figure 1C). In each selected household, all members aged 14 years and above were invited to participate in the study to assess within-household transmissions (Figure 1D). 

**Specimen collection and laboratory analyses.** During study visits at each household, fieldworkers collected venous blood samples using Ethylenediamine Tetraacetic Acid (EDTA) tubes from each consenting study participant 14 years and older. For ethical reasons, younger children could not provide venous blood samples at this stage. All laboratory methods are described in a previously published preprint [11]. In brief, we determined antibody reactivity using Anti-SARS-CoV-2-ELISA for IgG/IgA (Euroimmun Anti-S1-SARS-CoV-2-ELISA-IgG, hereafter EI-S1-IgG/Euroimmun Anti-S1-SARS-CoV-2-ELISA-IgA, hereafter EI-S1-IgA), and the Elecsys Anti-SARS-CoV-2 Roche anti-N pan-Ig (hereafter Ro-N-Ig). We also used the GenScript cPass assay. For serological confirmation, we used a virus micro-neutralization test as described previously [12]. To obtain accurate seroprevalences, we performed validation studies using a panel of 991 truly SARS-CoV-2-negative plasma samples from the pre-COVID-19 era and 193 samples from PCR-confirmed COVID-19 patients [11].

While the agreement between EI-S1-IgG and Ro-N-Ig assays was generally high (Online Appendix A), the latter gave more valid results. For this assay, an optimized cut-off of 0.4218 (instead of 1.0) yielded a sensitivity of 88.60% and a specificity of 99.72% [11]. We therefore used the Ro-N-Ig assay with this optimized cut-off to determine seropositivity for SARS-CoV-2 in our analyses. Robustness of the prevalence estimates was tested with EI-S1-IgG, EI-S1-IgA, and combinations of different assays as well as different cut-offs and measures of test performance to predict seropositivity (Online Appendix A).

**Household and personal data collection.** During household visits, field workers used the mobile data collection tool OpenDataKit (ODK) to capture contact details of household members on Android smartphones. Participants completed a household form and a personal questionnaire online using a newly developed web-based application. Non-responders were reminded by email no later than 2 weeks after the household visit, followed by telephone reminders. Telephone interviews were offered to those who felt unable to complete the questionnaires online.

**Statistical analyses.** All statistical analyses were performed using the statistical software R (version 4.0.2, R Development Core Team, 2020).

We calculated absolute and relative frequencies of sociodemographic and household variables and compared them to data of the general Munich population. Online and telephone responses were compared using Fisher’s exact test if there were only 2 categories and chi-squared test if there were more than 2 categories.

To assess the seroprevalence, we defined SARS-CoV-2 seropositivity on the basis of the Ro-N-Ig test result, applying the optimized cut-off of 0.4218 as described above. To account for the sampling design, we computed the sampling weights and calibrated them such that the sample structure mirrored the Munich population (regarding age, sex, migration background, presence of children in the households, single-member households). No spatial autocorrelation was assumed as the calculation of Moran’ I was not statistically significant. Therefore, in our different analyses, we did not account for the spatial autocorrelation (e.g., at the district level; Online Appendix A). Prevalence estimates were calculated using the calibrated weights, and 95% confidence intervals were computed on the basis of the variability associated with the sampling design (Online Appendix A). These prevalence estimates were additionally adjusted for sensitivity and specificity of the test as described by Sempos and colleagues (for details on sampling design and sensitivity/specificity adjustments, see Online Appendix A) [9]. We calculated the infection fatality ratio (IFR) for the population aged 14 years and older using the seroprevalence estimates for Munich as described above and officially reported numbers of COVID-19 deaths. Fatality counts follow a binomial distribution with small success probability parameter and relatively large number of trials, and thus can be approximately described by a Poisson distribution. This assumption leads to estimates of seroprevalence and corresponding 95% confidence intervals (CI), both for the entire period of our study as well as for weekly incidences. Official numbers include both subjects living in private households and institutions (e.g., homes for the elderly). To the best of our knowledge, there are no data for Munich on the percentages of SARS-CoV-2 infections and deaths occurring in institutions. For Germany, the Robert Koch Institute (RKI) reported that up to the end of the study period, 13% of infections and 46% of deaths occurred in institutions [13]. Given the lack of data for Munich, we estimated the IFR for Munich assuming that the percentage of deaths occurring in members of private households lay in the range between 20 and 100. Likewise, we calculated the factor of underreported infections assuming that between 20 and 100% of reported infections occurred outside institutions (Online Appendix A).

In the risk factor analyses, associations between the personal and household level covariates and seropositivity were evaluated using logistic regression, i.e., generalized linear models (GLM). We adjusted for age and sex and assumed item-nonresponse to be missing at random. The computed odds ratios (ORs) and 95% CIs were compared to the results of generalized linear mixed models (GLMM), which allowed us to consider the effect of household clustering during the estimation process [14]. For sensitivity analysis, we imputed missing values (of covariates) under the (Bayesian) joint analysis and imputation of incomplete data (JointAI) framework, which allowed us to avoid pooling [15]. The identified important risk factors for seropositivity were included in multiple regression models. Specifically, we compared the frequentist setup of GLM and GLMM as well as a GLMM using simultaneous imputation of multiple missing covariates under the Bayesian framework (Online Appendix A).

To analyze clustering of SARS-CoV-2 infections, we used the similarity of seropositivity levels within spatial clusters of different sizes, i.e., households, buildings, and geospatial clusters of various distances. As a test statistic, we employed the average within-cluster variance. To assess significance, we performed a non-parametric approximate permutation test with *n* = 10,000 randomly permuted measurement assignments [16]. To account for household clustering, when analyzing buildings and geospatial clusters, we only permuted households of the same size (Online Appendix A).

For official incidence and mortality, as well as for data on the general population, we used data provided by the Statistical Office of the City of Munich.

## 3. Results

**Description of the study samples and population.** Of the 6896 households identified, 4903 were eligible and 2994 were included in the analyses. Within these households, fieldworkers invited 6117 persons to participate, of which 5313 agreed and provided blood samples (Figure 2).

The study population was comparable to the Munich population with respect to sex (52% vs. 50% women) (Table 1). However, it contained less children and adolescents (5%) than the general population (17%) as children younger than 14 years had been excluded. In addition, persons born outside Germany were underrepresented (18% in the study population vs. 31% in the general population). Regarding household characteristics, the sampling design resulted in a preference for larger apartment buildings with 71% of the study population living in apartment houses with five or more apartments compared to 34% of the Munich population.

Some groups were more likely to participate in telephone interviews than to complete the online questionnaire—women (59% of telephone interviewees vs. 52% of online interviewees), participants aged 65+ years (66% vs. 12%), and subjects with a lower level of education (72% vs. 26%). These marked differences were also reflected in other personal characteristics such as employment status and smoking behavior, as well as in health-related-parameters (Online Appendix A).

**Prevalence of SARS-CoV-2-specific antibodies and mortality.** The highest number of PCR-positive individuals reported by the Statistical Office of the City of Munich was registered in March 2020 before the start of the fieldwork, reaching a maximum in week 14 (Figure 3A,D). The overall number of PCR-positive individuals 14 years and older from the beginning of March until the end of the fieldwork was 6293. This corresponded to a positivity rate of 0.46%. We began the fieldwork in week 15 and ended in week 24 (Figure 3B). Over these weeks, official case numbers went markedly down. In contrast, Ro-N-Ig seroprevalence stayed stable throughout the study period (Figure 3E and Online Appendix A). Using the optimized cut-off, the crude Ro-N-Ig seropositivity for the whole study population was 1.75% (95% CI 1.28–2.22%), with similar results when accounting for the sampling design using calibrated weights (1.89%; 95% CI 1.41–2.37%). Adjusting for test sensitivity and specificity slightly lowered the prevalence estimate (unweighted: 1.67%, CI 1.13–2.20%; weighted: 1.82%, CI 1.28–2.37%) (Online Appendix A). The estimated number of cases was about fourfold the official number (Figure 3C). Assuming (on the basis of data for the whole of Germany) that 87% of infections occurred in private households, we found that 4.5 times (95% CI 3.2–5.9) more infections than reported occurred (for details, see Online Appendix A).

On the basis of official mortality data for the Munich population, we observed an excess mortality in weeks 14 to 19, 2020, compared to the previous 4 years (Figure 3G,H). The resulting overall number of deaths over the whole period was similar to the 216 COVID-19 deaths reported by the Statistical Office of Munich (Figure 3I). Assuming the weighted and adjusted Ro-N-Ig prevalence of 1.82% for the Munich population 14 years and older (1,369,444 inhabitants), we estimated that until the end of the field work, 24,990 individuals developed SARS-CoV-2 antibodies (for details on this and the following calculations, see Online Appendix A). Up to the end of the fieldwork, the 216 registered COVID-19-related deaths yielded an IFR of 0.86% (95% CI: 0.67–1.23%) (Figure 3F). Estimating that only 54% of them occurred in private households, IFR lowered to 0.47% (95% CI: 0.36–0.67%).

**Associations with SARS-CoV-2 seropositivity.** Bivariate analyses suggested that Ro-N-Ig seropositivity depended only weakly on most surveyed factors. Loss of the sense of smell or taste at the time of the study was associated with the outcome; however, the confidence interval was wide (OR 41.3; 95% CI 6.7–231.0) using a classical GLMM. In addition, respiratory allergies (OR 3.3; 95% CI 1.1–10.3) were statistically significantly associated with Ro-N-Ig seropositivity. Only weakly related to the outcome were those working in a high-risk job, household type, and living area per inhabitant (Figure 4 and Online Appendix A). Besides the loss of the sense of smell or taste, which was considered a symptom of the outcome rather than a risk factor, we included these variables in the final GLMM in which none of the associations were statistically significant. When we applied Bayesian GLMM with imputation of missing values in the sensitivity analyses, we obtained similar results (Figure 5 and Online Appendix A).

**SARS-CoV-2 transmission within households, buildings, and neighborhoods.** Ro-N-Ig test outcomes of participants had a significantly lower variance within households than among the entire population (Figure 6 and Online Appendix A). In contrast, we did not find a statistically lower variance within buildings (*p* = 0.26) nor within neighborhoods applying radii from 50 (*p* = 0.16) to 4000 m (*p* = 0.78). Yet, a lower-than-expected mean variance was seen up to a distance of 200 m.

## 4. Discussion

We present an estimate for the SARS-CoV-2 seroprevalence in the Munich general population 14 years and older, which was still low towards the end of the first pandemic wave (1.82%). However, our results indicate that the seroprevalence was substantially higher than official numbers in terms of registered PCR-positive cases. We could only identify weak risk factors for SARS-CoV-2 seropositivity. Finally, our data confirmed household clustering of infection [17].

As study participants were enrolled at a time when the newly released serological assays were not fully validated, we carefully evaluated three primary serological assays [11]. As similarly reported by Gudbjartsson et al., the specificity and sensitivity of Roche anti-N pan-Ig was superior to Euroimmun [6]. By using the cut-off index raw values, we were able to optimize the cut-off of the assay.

Our prevalence estimates are in line with findings from Gudbjartsson et al. who estimated the SARS-CoV-2 antibody seropositivity for the general population of Iceland at around 1%, thus being slightly lower than in our population [6]. Other studies estimating seroprevalence for European general populations reported results between 2% in Luxembourg [18]; 4% in Spain [19]; and 11% in Geneva, Switzerland [20]. The proportion of officially registered cases vs. the number of serologically positive cases also varied significantly between the studies. In our study, about one in four to five seropositive cases were officially registered, although one needs to consider that we did not have data on the seroprevalence among children younger than 14 years. For these, only one out of six infections might have been officially registered, as recently shown by Hippich et al. [21] for children living in Bavaria, Germany. In addition, our study population did not include institutions in which residents might be more frequently tested than in other settings. Therefore, our estimate might even be conservative for the Munich overall population. In other population-based European studies, 10–56% of infections were detected by the healthcare system [6,19,20]. We believe that the proportion of registered vs. seropositive but not registered individuals could be used to measure the efficiency of a public health testing system.

While only a small proportion of the general population in Germany live in institutions, official data of the Robert Koch Institute (RKI) report that during the study period, 13% of reported SARS-CoV-2 infections and 46% of COVID-19-related deaths occurred in institutionalized persons [13]. Therefore, we report a range of IFRs depending on the percentage of deaths occurring in private households. Translating the RKI numbers to Munich, the IFR resulting from our study was 0.47% (95% CI 0.36–0.67%). IFR calculations are difficult to compare for the reasons mentioned above and, e.g., due to different approaches or unequal case ascertainment. The IFR resulting from our study is in line with recent data from Geneva, Switzerland (0.64; 95% CI 0.38–0.98%), but slightly lower than data reported for Spain (0.83; 95% 0.78–0.89) [22,23].

Among the risk factors identified, albeit not statistically significant, the increased risk of working in a job with a high potential of contact to COVID-19 cases appears plausible, and the result is in line with other studies [6,19,24]. Among the COVID-19 cases reported globally to the World Health Organization, 14% belong to the group of healthcare workers, whereas this group represents less than 3% of the general population in most countries [25]. Special preventive efforts should therefore be always targeting the group of workers occupationally exposed to SARS-CoV-2. We saw slightly increased odds of seropositivity among participants with respiratory co-morbidity, especially patients with allergies. Angiotensin-converting enzyme 2 (ACE2) receptors are thought to be main cell entries for SARS-CoV-2. Differences in expression levels of ACE2 in patients with allergic asthma might be one reason for the increased risk of SARS-CoV-2 seropositivity in our study population [26]. However, our data on respiratory allergies were self-reported, and therefore specificity might be low.

Participants with SARS-CoV-2 antibody responses more frequently reported the loss of the sense of smell or taste, even though the questionnaire was answered up to several weeks after the blood samples, which is consistent with other studies [27]. The underlying pathological mechanism might again be explained by the high concentration of ACE2 in olfactory cells [27].

Household clustering was already described for SARS-CoV-2 and is well known from other respiratory infections [17,24,28,29]. Apart from transmission from one person to the other within one household, similar behavior and contacts outside their own household might explain this. Our results weakly suggest clustering within the same house and eventually the close neighborhood. However, these associations were not statistically significant and need to be confirmed. With respect to within-household transmission, our seroprevalence observations are limited by the fact that we did not take any blood samples from children below 14 years. Since the number of children in a household is presumably positively correlated with household size, this biases our data, and within-household transmission may even be more pronounced than reported here.

The major strengths of this study are its population-based approach, weighting of results for the general Munich population, the high number of participants, the thorough validation of the assays used, and (where available) the use of validated questionnaire items. The overall response to the study was high compared to other population-based epidemiologic studies in Germany [30]. While most participants completed the questionnaire online, we also provided the alternative of telephone interviews, increasing participation and, thus, making our study population more representative for the target population. Our study started towards the end of the first pandemic wave in Munich, as illustrated by the incidence of SARS-CoV-2 registered cases and mortality. This resulted in a relatively stable antibody prevalence over the course of the fieldwork.

As mentioned, a relevant limitation of our study is the exclusion of children and residents not living in private households. While in general, people with migration background are less likely to participate in population-based studies, the lack of translated questionnaires further limited the number of migrants participating in our study [10]. To increase response, blood samples were collected at participants’ homes and not at a centralized testing facility. However, we could not interview participants during home visits. This led to lower questionnaire response and a time interval of up to six weeks between blood sample collection and completion of the questionnaire. Item non-response in online completed questionnaires was higher compared to personal interviews. As in other studies, questions on income (21% missing responses) as well as weight and height (73% missing responses) were especially prone to non-response. They were thus not included in the final models. Despite the high number of participants, the power for our risk factor analyses was relatively low due to the low seroprevalence, resulting in large confidence intervals and potentially negative findings. However, one may assume that any strong risk factors would have been identified if included.

Although commercial and scalable testing methods for SARS-CoV-2 seem robust, cut-offs have to be adapted to specific populations to improve their validity. However, the relevance of antibody-based testing to assess previous infection remains partly unclear. To learn more about the meaning of detected SARS-CoV-2-specific antibodies, correlates of protection, and testing methodologies, this as well as other longitudinal cohort studies need to be continued.

## 5. Conclusions

Even when considering all inherent uncertainties, our results indicate that only a very small proportion of the Munich population encountered SARS-CoV-2 during spring 2020, most citizens stay vulnerable to infection, and the associated risk of death among those infected was high. The careful evaluation of the serum samples contributed to the validity of our results. Methodologically, it provides insights into how a sound population-based epidemiologic study can be conducted early on in a pandemic. Finally, it is the only population-based study outside a hotspot thus far published in Germany, a country in which early public health measures possibly prevented the healthcare system from collapsing. We therefore consider the data relevant and useful to consider for healthcare providers, also for future pandemics. These results will help authorities to estimate the true burden of disease in the population and to make evidence-based decisions on public health measures.

## Figures and Tables

**Figure 1 ijerph-18-03572-f001:**
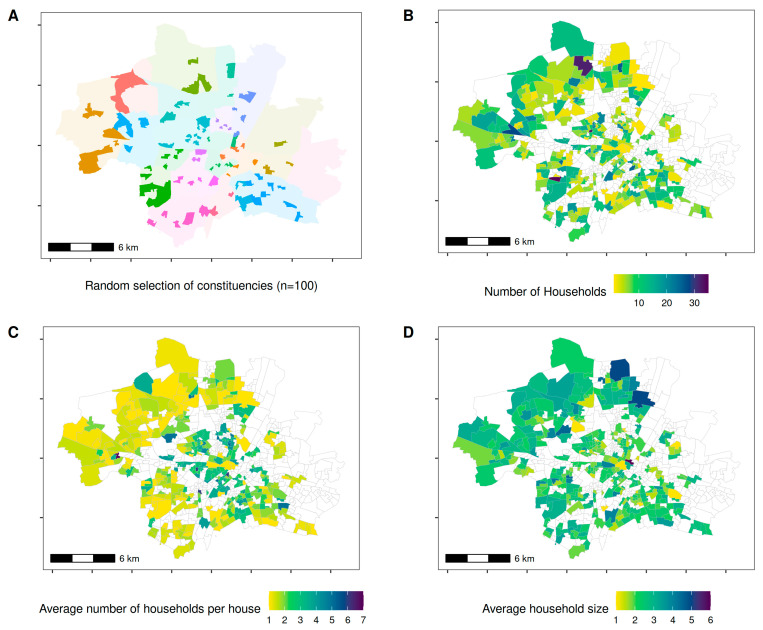
Selection procedure and geospatial distribution of the study population. (**A**) The municipality of Munich together with its districts (distinguished by different colors). The 100 selected start constituencies for the random walks are marked in the same color as the respective constituency but in a darker shade. (**B**) All 2994 included households and their respective 368 constituencies. (**C**) Average number of recruited households per building by constituency. (**D**) Average number of members per recruited household by constituency.

**Figure 2 ijerph-18-03572-f002:**
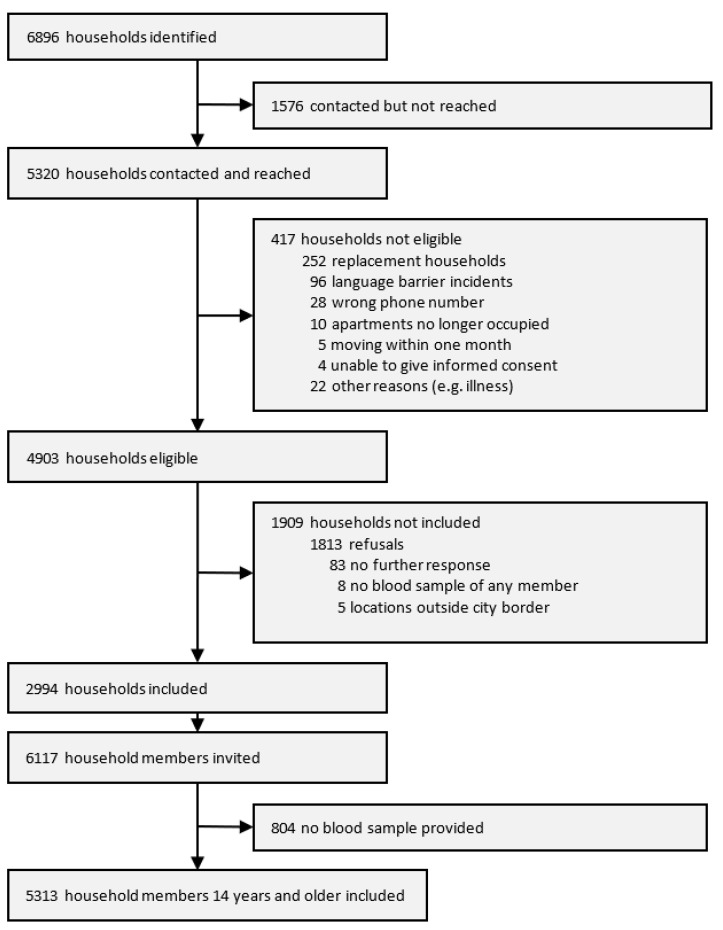
Flow chart on participant selection for the KoCo19 baseline survey.

**Figure 3 ijerph-18-03572-f003:**
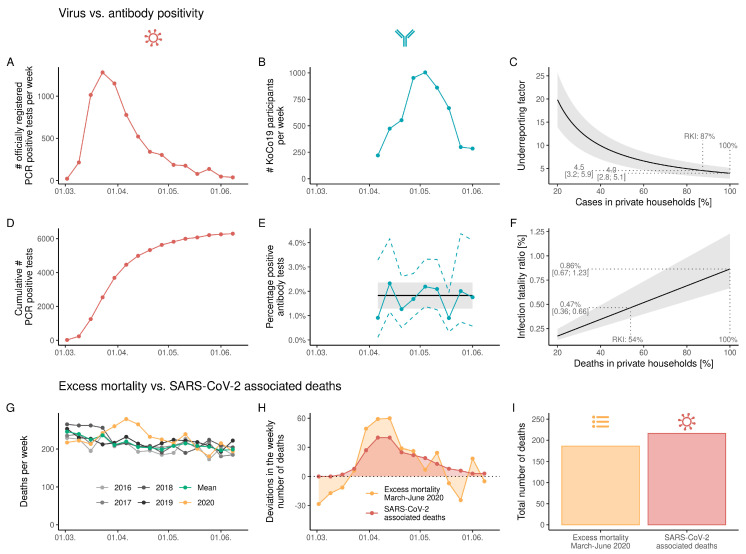
Dynamics of the COVID-19 pandemic and of the KoCo19 study in Munich since the beginning of the pandemic to the end of the KoCo19 study period. (**A**) Official weekly absolute number of newly diagnosed COVID-19 cases based on positive PCR tests. (**B**) Weekly number of participants recruited to the KoCo19 study. (**C**) Estimated underreporting factor depending on the percentage of reported cases in private households with respect to all reported cases in Munich. (**D**) Cumulative weekly number of officially registered COVID-19 infections in Munich. (**E**) Numbers of Elecsys Anti-SARS-CoV-2 Roche anti-N pan-Ig (Ro-N-Ig) seropositive samples per week (blue) divided by the number of blood draws in the respective time frame. 95% CIs (blue dashed lines) are based on an approximate Poisson assumption. Black line and shaded area indicate the weighted and adjusted prevalence estimate with 95% CI. Due to low recruitment numbers in the last week, in (**D**,**E**), the data from the last week were integrated with the pre-last week. (**F**) Estimated infection fatality ratio depending on the percentage of reported COVID-19-related deaths in private households with respect to all reported COVID-19-related deaths in Munich. (**G**) Weekly number of deaths in Munich for 2016–2020 in terms of official numbers. (**H**) Weekly excess mortality in 2020 compared to 2016–2019 in terms of official death counts and official SARS-CoV-2-related deaths. (**I**) Comparison of total number of deaths in terms of excess mortality and registered SARS-CoV-2-related deaths.

**Figure 4 ijerph-18-03572-f004:**
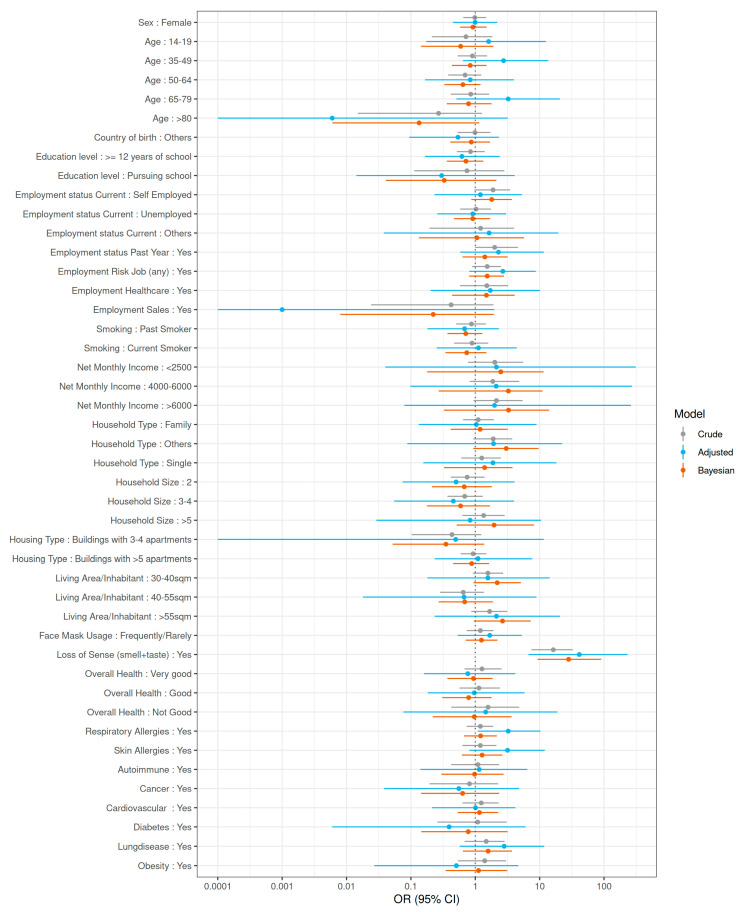
Risk factor analysis for SARS¬CoV-2 seropositivity. Risk factor analysis for SARS¬CoV-2 seropositivity in the KoCo19 study population comparing crude, adjusted for clustering, and Bayesian (after imputation and adjusted for clustering) estimates. All odds ratios (ORs) and 95% CIs were adjusted for age (continuous scale) and sex. OR: odds ratio; 95% CI: 95% confidence interval (frequentist GLMM)/95% credible interval (Bayesian analyses).

**Figure 5 ijerph-18-03572-f005:**
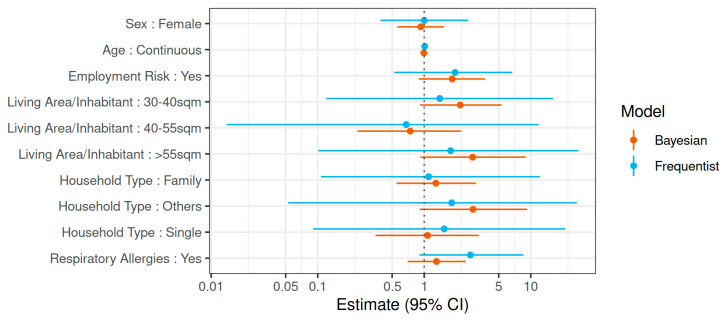
Multivariate risk factor analysis for SARS¬CoV-2 seropositivity. Multivariate risk factor analysis for SARS-CoV-2 seropositivity mutually adjusted for all variables in the figure. OR: odds ratio; 95% CI: 95% credible interval (Bayesian analyses)/95% confidence interval (frequentist GLMM).

**Figure 6 ijerph-18-03572-f006:**
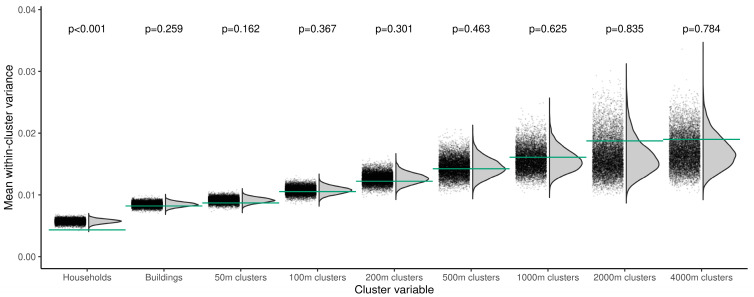
Proximity clustering of Ro-N-Ig test outcomes. We subdivide the participants into disjoint clusters according to various cluster definitions: households, buildings, and spatial clusters of various diameters (*x*-axis). For each cluster, we calculated the within-cluster variance of observed Ro-N-Ig test outcomes of all participants in the cluster. Their means over all clusters are marked by green horizontal lines for each cluster size. We then performed 10,000 random permutations of measurements assignments. The black dots show the respective mean within-cluster variances, along with density estimates as grey curves. For buildings and spatial clusters, measurements of a household were only permuted with measurements of a household of the same size. *p*-values indicate the one-sided probability of a random value being smaller than or equal to the observed one.

**Table 1 ijerph-18-03572-t001:** Individual and household characteristics of the KoCo19 study participants compared to the Munich population.

Characteristics	Munich Population		KoCo19 Study Participants		
	**Individual Characteristics**				
**N**	**1,561,720**		**5313**		
	**n**	**%**	**n_missing_**	**n**	**%**
**Sex**			0		
Female	789,437	50.1		2766	52.1
**Age (years)**			0		
0–19	263,053	16.8		267	5.0
20–34	390,382	25.0		1346	25.3
35–49	348,651	22.3		1542	29.0
50–64	291,562	18.7		1306	24.6
65–79	184,764	11.8		676	12.7
80+	83,308	5.3		176	3.3
**Country of birth**			465		
Outside Germany	476,575	30.5		849	17.5
**Level of education**	NA	NA	701		
Still in school				100	2.2
<12 y				1386	30.1
≥12 y				3126	67.8
**Employment status**	NA	NA	576		
Employed				2911	61.5
Self-employed				471	9.9
Not working ^1^				1258	26.6
Others ^2^				97	2.0
**Risk employment ^3^**	NA	NA	470		
Yes				851	17.6
	Household characteristics				
***N***			2994		
**Housing type: building with**	148,607	100	0		
1–2 apartments	82,119	55.3		661	22.1
3–4 apartments	10,938	7.4		192	6.4
≥5 apartments	50,339	33.9		2137	71.4
Others ^4^	5211	3.5		4	0.1
**Household type**	854,288	100	307		
Single	468,937	54.9		680	25.3
Couple	160,339	18.8		922	34.3
Family	185,752	21.7		875	32.6
Others ^5^	39,260	4.6		210	7.8
**No. of household members**	854,288	100	1		
1	468,937	54·9		784	26.2
2	193,376	22·6		1171	39.1
3–4	106,074	12·4		880	29.4
5+	85,901	10.1		158	5.3
**Living area per inhabitant**	NA	NA	319		
≤30 m^2^				800	29.9
30–40 m^2^				634	23.7
40–55 m^2^				579	21.6
>55 m^2^				662	24.7
**Net family income**	NA	NA	924		
EUR ≤ 2500				445	21.5
EUR 2500–4000				502	24.3
EUR 4000–6000				607	29.3
EUR > 6000+				516	24.9

^1^ “Not working” includes unemployed, retired, parental leave, sabbatical, students; ^2^ “others” includes voluntary social year, military service, part-time jobber, reduced working hours; ^3^ considered as “risk employment” for COVID-19 infections were employees in the healthcare sector, emergency service, senior homes, airport, public transport, education, sales, social work, and other risk jobs; ^4^ other types of housing include tents, caravans, or the like; ^5^ other household types include shared apartments by, e.g., students, subleasing, and assisted accommodation.

## Data Availability

Our data are accessible to researchers upon reasonable request to the corresponding author taking data protection laws and privacy of study participants into account.

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
