# Peer review of "Prevalence and Risk Factors of Infection in the Representative COVID-19 Cohort Munich"

_ijerph, 2021, doi:10.3390/ijerph18073572_

Round 1

Reviewer 1 Report

Generally speaking, the study is very well written, I would only suggest making few changes. The study is fairly interesting and the information is also well organised. Regarding that it is an epidemiological study, the authors have considered a high volume of samples, which makes the study a robust approach. The results reported here are also essential in order to understand the dynamic of the infection, especially in a densely-populated area. However, due to the high dissemination of the COVID-19 and the subsequent volume of data acquired worldwide, these outcomes should have been published before. As a consequence, this report is now obsolete. The results need to be deeply discussed or it is necessary to update the information in order to provide a tool to the system care. If the dynamic of the infection has changed from June 2020 to now, this also should be discussed. If not, the report should mention it.

By no means is it necessary to make a major revision, however, I would like to read the author´s response about what I am referring here.

Minor English and editing changes

(ii) to identify risk factors for SARS-CoV-2 seropositivity, and (iii) to  assess household and neighbourhood transmission patterns.

Figure 1. Unless the reader lives in Munich, it is difficult to know the real dimension of the city. So, I recommend adding a scale at the top or bottom of each figure just to have a better understanding of the study.

Household clustering of SARS-CoV-2 was described before and is well-known

…outside their own household might explain this.

As mentioned above or As it was mentioned before,

Reviewer 2 Report

The abstract does not evidence in an effective way the main aims of the study and it does not seem very "attractive" for readers. I suggest to better evidence the possible added value respect to the actual state of the art and to put in evidence the purposes of the work.

According to me, the introduction should provide a wider contextualization, also with the support of other literary review about the situation in Munich, and in Germany too, and better explain the main aims of the paper and its possible added value for the scientific state of the art.

At the beginning of par. 2 there is written "A detailed description of the study design, setting and population was previously published [10]." Some other information are required to better explain the main characteristics of study design and popluation involved.

In fig. 1A which is the means of the colours? Please specify in legend.

About the following "We calculated absolute and relative frequencies of sociodemographic and household variables and compared them to data of the general Munich population". Please provide in-depth information about sociodemographic and household variables which have been calculated.

About "The study population was comparable to the Munich population in many aspects (Table 1)."  Please provide a synthesis of comparable characteristics because at the moment there is only a reference to table 1.

About "Among the risk factors identified, albeit not statistically significant, the increased risk of working in a job with a high potential of contact to COVID-19 cases appears plausible and is in line with other studies". It appears obvious. Please specify something about this affirmation and provide some specific examples to add critical considerations and an useful framework.

Which are the information required in the questionnaire? A table regarding the information required and a graphs with the synthesis of the main aggregated results colud be useful.

Are there any findings in terms of spatial distribution and analysis? Please, if it is the case, provide some information on this issue.

The conclusions are very short and limited and they do not provide an added value to the scientific discussion. Please provide i.e. a brief propositional synthesis of the main results obtained in the perspective of possible future researches which could have benefits by this work.

Reviewer 3 Report

The study does not bring anything revealing to the subject of COVID-19

Round 2

Reviewer 1 Report

The rebuttal letter has provided sufficient information with respect to what I requested. It is now clear that this publication would mean an interesting background to better understand the behaviour of highly-density populations not only in this specific pandemic but also in future similar situations.

Reviewer 2 Report

I think that this paper, which is focussed on a very important and actual theme, has recorded a good improvment by the revisions that the Authors have done.

Reviewer 3 Report

I still believe that the work does not bring anything innovative that is currently available